# Language-based EMA assessments help understand problematic alcohol consumption

August Håkan Nilsson [1,2]*, Hansen Andrew Schwartz[3], Richard N. Rosenthal[4], James R. McKay[5], Huy Vu[3], Young-Min Cho[1], Syeda Mahwish[3], Adithya V. Ganesan[3], Lyle Ungar[1]

**1** Department of Computer and Information Science, University of Pennsylvania, Philadelphia, Pennsylvania, United States of America, **2** Oslo Business School, Oslo Metropolitan University, Oslo, Norway, **3** Department of Computer Science, Stony Brook University, Stony Brook, New York, United States of America, **4** Department of Psychiatry, Renaissance School of Medicine at Stony Brook University, Stony Brook, NY, United States of America, **5** Department of Psychiatry, University of Pennsylvania, Philadelphia, Pennsylvania, United States of America

\* august.nilsson1907@gmail.com

## Abstract

### Background

Unhealthy alcohol consumption is a severe public health problem. But low to moderate alcohol consumption is associated with high subjective well-being, possibly because alcohol is commonly consumed socially together with friends, who often are important for subjective well-being. Disentangling the health and social complexities of alcohol behavior has been difficult using traditional rating scales with cross-section designs. We aim to better understand these complexities by examining individuals' everyday affective subjective well-being language, in addition to rating scales, and via both between- and within-person designs across multiple weeks.

### Method

We used daily language and ecological momentary assessment on 908 US restaurant workers (12692 days) over two-week intervals. Participants were asked up to three times a day to "describe your current feelings", rate their emotions, and report their alcohol behavior in the past 24 hours, including if they were drinking alone or with others.

### Results

Both between and within individuals, language-based subjective well-being predicted alcohol behavior more accurately than corresponding rating scales. Individuals self-reported being happier on days when drinking more, with language characteristic of these days predominantly describing socializing with friends. Between individuals (over several weeks), subjective well-being correlated much more negatively with drinking alone ($r = -.29$) than it did with total drinking ($r = -.10$). Aligned with this, people who drank more alone generally described their feelings as *sad*, *stressed* and *anxious* and drinking alone days related to *nervous* and *annoyed* language as well as a lower reported subjective well-being.

**Data Availability Statement:** The numerical data underlying the analyses, as well as the code for analyzing these are available at Open Science Framework, here: https://osf.io/tgvh5/. Due to

sensitivity, we can not share the language data publicly. The language data may be available on request. Contact: Leo Thorbecke, Research Project Manager at the University of Pennsylvania's schools of engineering and medicine. Email: Leo. Thorbecke@pennmedicine.upenn.edu. Leo will be able to grant access to the language data upon certain requests.

**Funding:** Initials of the authors who received each award: Ungar, L., Schwartz, A. Grant numbers awarded to each author: 1R01AA028032-01 The full name of each funder. National Institute on Alcohol Abuse and Alcoholism URL of each funder website: https://www.niaaa.nih.gov/ The funders had no role in study design, data collection and analysis, decision to publish, or preparation of the manuscript.

**Competing interests:** The authors have declared that no competing interests exist.

## Conclusions

Individuals' daily subjective well-being, as measured via language, in part, explained the social aspects of alcohol drinking. Further, being alone explained this relationship, such that drinking alone was associated with lower subjective well-being.

## Introduction

Alcohol consumption is a severe public health problem where 4% of the global disease burden is attributable to alcohol [1]. It can lead to morbidity and mortality [2] and relates to common mental disorders [3]. Yet, alcohol consumption often occurs in social settings [4], and social connectivity, including friendships, serves as one of the strongest predictors of subjective well-being (SWB; [5–7]) and physical health [8–10]. Recent meta-analyses show that individuals, including those in clinical populations, in fact drink more on the days they feel more positive emotions and not when they feel more negative emotions [11, 12]. However, meta-analyses of studies of laboratory inductions of negative affect (as against controls) demonstrate small-to-medium effects on increasing alcohol use [13, 14] and craving [13]. In addition to affective valence contributing to the probability of increased alcohol intake, emotion regulation strategy, and even more so, behavioral aspects of negative emotion regulation ability (e.g., impulsivity), appear to have a key role in the initiation of substance use, including alcohol across both positive and negative emotions [15]. Thus, alcohol consumption is a complex behavior, and although it mostly relates negatively to physical health, non-problematic drinking can have a positive relationship to mental health and SWB through its strong social component [4].

We aimed to learn more about psychologically *problematic* alcohol consumption by analyzing language assessments of individuals' described feelings, i.e., *language assessments of momentary, affective SWB*, in addition to traditional SWB rating scales. Thus, we focused on the psychological component and not the physical component of alcohol consumption. We used ecological momentary assessment (EMA) with restaurant workers, a large population with higher than usual problematic alcohol drinking [16], with a particularly risky work environment involving high job stress, low job autonomy, shift work, and a drinking culture in the workplace [1, 17]. We analyzed their drinking behavior in relation to both their language-assessed SWB and their rated SWB. We compared whether the language assessments analyzed with state-of-the-art natural language processing predicted alcohol consumption more accurately than the traditional rating scales, both within individuals on a daily basis and between individuals across multiple weeks. By determining the specific affective language that predicted drinking behavior and if the drinking occurred alone or with others, we were able to better understand the mechanisms behind the relationship between SWB and alcohol drinking.

### Drinking and subjective well-being

SWB is an important end goal for most people, according to several researchers [18–20], philosophers [21], and individuals [22, 23], and composes two dimensions. The affective component (also called experienced well-being) involves the day-to-day emotions that individuals experience [24, 25], and the cognitive, evaluative component involves the overall evaluation of one's life and is often operationalized as life satisfaction [26] or harmony in life [27]. Research on the relationship between alcohol and SWB often involves either studying how alcohol relates to daily fluctuations in affect [11, 12] or the cognitive, evaluative component of SWB

[28–30]. Understanding both components, together and individually, in relation to alcohol consumption may help to understand when and how alcohol consumption can be problematic. Since SWB is highly valued among most people [22, 23] and positively relates to physical health [18], it may be used to understand problematic alcohol use.

Alcohol consumption has a mixed relationship to SWB's two components (i.e., the cognitive, evaluative component and the affective component). The days when people drink more are associated with higher affective SWB. A recent meta-analysis analyzed 69 EMA studies on daily reported drinking and affective SWB in community, college, and clinical samples and found that all these groups drink more on days when they feel more positive affect and not when they feel more negative affect [11]. At the pre-registration stage of the meta-analysis, more than half of the 38 authors of the meta-analysis reported in a survey that they expected drinking to be higher on days with more negative affect. They concluded that the prior beliefs regarding drinking and affect should considerably shift towards expecting drinking to be preceded by positive affect and not negative affect [11].

Life Satisfaction, often representing the cognitive, evaluative component of SWB, has a hump-shaped relationship to alcohol consumption where low to moderate drinkers are the most satisfied with their overall life and heavy drinkers the least satisfied [28–30]. Crucially, increased alcohol consumption to a point has been shown to not lead to lower life satisfaction in general, but specifically, entering heavy alcohol consumption and increases in alcohol problems have significantly decreased life satisfaction [28]. High consumption of alcohol may be associated with an Alcohol Use Disorder (AUD), which relates to and causes negative consequences of alcohol, such as morbidity [31] and decreased psychosocial functioning [32]. Taken together, alcohol consumption is not necessarily problematic for SWB in the general population, but heavy drinking and drinking problems are problematic both by reducing SWB and harming physical health.

## Social aspects of drinking

The degree to which drinking is a social versus solitary activity might help explain when drinking is negative for SWB. Most drinking occurs socially [33], but 27% of adolescents and 40% of young adults report, in a 4000+ US nationally representative sample, they have been drinking alone at least once during the past 12 months. Spending time with others is strongly associated with reporting high levels of positive affect and life satisfaction [34, 35], and individuals high in SWB spend less time alone [6]: in one study sample, everyone in the top decile had at least one close relationship [6]. Conversely, loneliness is a strong predictor of poor SWB and physical health [36]. It is thus relevant to disentangle social and solitary drinking when considering problematic alcohol consumption.

Those who report solitary alcohol use appear to do so primarily to cope with negative emotions [4, 37–41], a pattern of alcohol use that has been consistently linked to the development of alcohol problems [4, 42, 43]. Meta-analytic results show that drinking alone predicts alcohol problems cross-sectionally, although the effect size is often weak [40]. However, there is evidence that solitary drinking relates to alcohol problems indirectly through the coping-with-negative-emotions motives [44]. Longitudinally, the relative frequency of drinking alone in adolescence predicts alcohol use problems and AUD symptoms at age 25 independently of baseline alcohol use and problems in both community and clinical samples [45]. A recently published 17-year longitudinal study supported the effects of solitary drinking in adolescence on AUD symptoms at age 35, controlling for other risk factors of alcohol usage [33]. Thus, drinking alone seems to be an early sign of problematic alcohol consumption. The sparse evidence on drinking alone versus with others with EMA data suggest that positive and negative

affect are lower and higher when drinking alone, respectively [46], that sedation is higher when drinking alone, and that the pleasure of drinking reduces faster when drinking alone [47]. In sum, drinking alone might be an important factor that distinguishes *problematic* alcohol consumption from non-problematic drinking and predicts lower SWB.

## Using natural language to understand alcohol consumption

The natural language individuals use contains information beyond the numbers derived from rating scales. Asking people to describe their feelings open-endedly (e.g., "How are you?") is seemingly more ecologically valid than asking them to rate their emotions on a corresponding rating scale (e.g., "How are you on a scale from 1 to 10?"). Answers to open-ended questions about individuals' evaluative SWB, both satisfaction with life and harmony in life, have predicted corresponding rating scales at $r = .80$ and $r = .85$, respectively [48], Harmony in life descriptions have significantly predicted cooperative behavior, which the corresponding rating scale did not [49], and Facebook status updates predicted high-risk AUD individuals with superior performance to a stress survey scale [51]. Besides the predictive accuracy, language can provide unique psychological insights by revealing the specific language that is associated with an outcome. For example, high-risk drinkers have been shown to use more *informal*, *party*, and *sexual* language than low-risk drinkers in their Facebook updates, who, in turn, use more *religious* and *social* language [50].

In this study, we used individuals' EMA-based SWB language by asking the participants to describe their current feelings every day over multiple weeks. First, by analyzing the *within-person* variation in language and drinking, we could understand the language that individuals use more on days when drinking more and when drinking alone. Second, by analyzing the *between-person* variation of the aggregated language from each person's total EMA responses, we could have a noise-reduced assessment of the affective SWB language that relates to individuals who drink more alone and drink more in general.

To predict alcohol behavior from language, we used a state-of-the-art transformers-based Large Language Model to convert the language into numerical vectors (called *word embeddings*). These word embeddings were trained to predict alcohol behavior. Transformers-based large language models can contextualize words such that they understand the different meanings of "bank" in "bank account" and "river bank" and have outperformed other language analyses in predictive accuracy [51]. We hypothesized that the affective SWB language would be more accurate than rating scales in predicting drinking behavior.

To understand the specific language driving these predictions of alcohol consumption, we created data-driven topics of the language through the algorithm Latent Dirichlet Allocation [52]. This algorithm clusters groups of words occurring together (becoming topics), and we then correlate the relative frequency of the topics with alcohol consumption. All the topics can be associated with an outcome, and thus, both topics that relate to high and low SWB can be related to drinking outcomes. This is particularly relevant for understanding alcohol consumption since individuals drink for different reasons and because of the hump-shaped relationship drinking has to Life Satisfaction [28–30].

In sum, the study aims involved the following:

1. To answer the research question, "Does language-based affective SWB predict drinking behavior more accurately than rated SWB?". We hypothesized that the language-based SWB would be more accurate.

2. We asked, "What is more negatively related to SWB, drinking or drinking alone?" where we hypothesized that drinking alone would be more negatively related to SWB.

3. We aimed to answer the research question "What specific language relates to drinking behavior?".

## Method

### Ethical statement

The study is part of a larger study with the working title, "Data Science for Unhealthy Drinking (DS4UD)," H Andrew Schwartz, PI. The study received its ethical approval from the University of Pennsylvania Institutional Review Board. All participants provided digital informed consent to participate in the study. In the consent information, they were provided information specific to the study design and the details of their commitment, along with contact information for study personnel. Additionally, participants were told about their right to withdraw from the study at any given moment and told that the data would only be analyzed at the group level and no sensitive data would be made public.

### Procedure

US restaurant workers were invited to participate in a study of smartphone-delivered EMA as part of a larger study with the working title, "Data Science for Unhealthy Drinking (DS4UD)," H Andrew Schwartz, PI. Participants were recruited between June 2020 and June 2021 from different service care organizations and snowball sampling on social media. They signed up and consented at Qualtrics, where they were directed to download an app designed for this study to use for the EMA. To date, the study involved a total of three waves of two weeks each, the first collected in 2021 and the second and third in 2022.

At the beginning of each wave, new participants filled in an initial baseline survey, and recurrent participants took a wave survey with the same questions as the baseline except for questions about stable traits, such as sex. Participants were randomly assigned (50/50) to report EMA once or thrice daily during the two weeks after the baseline/wave survey. In each EMA response, participants were asked how many units of alcohol they drank during the past 24 hours, accompanied by a description of the units. In the last two waves, participants also indicated whether they had been drinking alone or with others, whenever they had been drinking. When a participant reported the number of drinks several times a day, we used the first reported number of drinks for each day of the participants to capture the drinking of the past day. After the alcohol drinking questions, participants were asked about their SWB through, first, the language assessment and second, the rating scales, which are described in the instruments section.

The data collection procedure was made such that the participants were principally non-identifiable to the authors. However, the text data participants provided at times included sensitive personal data. The data was kept in remote machines with restricted access, and only some of the researchers from the larger project had access to the sensitive language data.

### Participants and EMA days

There were 1175 participants that reported at least one EMA day, summing up to 24465 EMA responses over 13179 EMA days. To reduce noise, we excluded all participants who reported less than four EMA days, which yielded a sample size of 908 with a total of 12692 EMA days, or 14 days per participant on average. Most of the 908 participants partook only in the first wave (66%), a fourth partook in two or all three waves (26%), and some participants only partook in the second or final third wave (8%). We analyzed both *drinking yesterday* and *drinking tonight*. Since the original drinking question asked participants how much they drank in the

past 24 hours, we used the language of day n with the reported drinks of day n+1 to analyze drinking tonight. When analyzing drinking tonight, a total of 10171 EMA days remained. The final sample consisted predominantly of females (75%), had a mean age of 37 (SD = 8), and were all US restaurant workers. The participants were paid 1.50 US dollars per EMA and could, as a maximum, receive 113 US dollars per wave if they partook in all EMAs.

## Materials

**Baseline.** The baseline survey assessed demographic measures, such as age and gender. It also included the following measures:

*Risky alcohol consumption.* This was assessed by The Alcohol Use Disorder Identification Test (AUDIT). The 10-item AUDIT [53] assesses risky alcohol consumption and is widely used in clinical settings. Items include, for example, "How often do you have six or more drinks on one occasion?" and "How often do you have a drink containing alcohol?". The scale had a Cronbach's alpha of .88, .89, and 87 and McDonalds Omega of .90, 91, and .92 for waves 1, 2, and 3, respectively. The test-retest reliability was *r* = .72 between waves 1 and 2 (only new participants answered the 10-item AUDIT in wave 3).

**Affect.** We assessed affective SWB at the baseline and survey level by the 10-item short version of the Positive and Negative Affect Schedule [25, 54]. The items in this measure ask how much each respondent felt different emotions, such as "active", "inspired", and "upset," during a specific time period. In this study, we asked about the affect during the past two weeks. The positive affect scale had a Cronbach's alpha of .82, .83, and .74 and McDonald's Omega of .85, 87, and .83 for waves 1, 2, and 3, respectively. The negative affect scale had a Cronbach's alpha of .81, 83, and 77 and McDonald's Omega of .85, 87, and .90 for waves 1, 2, and 3, respectively. The test-retest reliability was *r* = .44 and *r* = .53 between waves 1 and 2 for positive and negative affect, respectively (only new participants answered the PANAS in wave 3).

*Life satisfaction.* For evaluative well-being, we used the single-item Cantril Ladder [55], which measures Life Satisfaction by asking respondents to place themselves on a ladder representing their worst (bottom) to best (top) possible life. The Cantril Ladder is used in many settings, including the Gallup World Poll, whose data forms the World Happiness Report. The scale had a test-retest reliability of *r* = .66 between waves 2 and 3 (the scale was not used in wave 1).

*Depression.* We assessed depression by the Patient Health Questionnaire-9 for Depression [56]. This scale involves 9 items and is often used clinically and involves answering how much one has experienced items such as "Feeling down, depressed, or hopeless" and "Feeling tired or having little energy". The scale had a Cronbach's alpha of .90 for waves 1 and 2 and .91 for waves 3, and the McDonalds Omega was .93 in all three waves. The test-retest reliability was *r* = .74 between waves 1 and 2 and *r* = .67 between waves 2 and 3.

*Loneliness.* We measured perceived loneliness by the revised UCLA scale [57]. The scale includes three items, "How often do you feel that you lack companionship?", "How often do you feel left out?" and "How often do you feel isolated from others?". The scale had a Cronbach's alpha of .84 and .77 and McDonald's Omega of .85 and .78 for waves 2 and 3, respectively. The scale had a test-retest reliability of *r* = .39 between waves 2 and 3 (the scale was not used in wave 1).

**Ecological momentary assessment.** *Language assessments of momentary, affective subjective well-being.* We had two language assessment questions of momentary, affective SWB. The first, in the form of a short essay, always asked participants, "Using the box below, please describe in 2 to 3 sentences how you are currently feeling." with a minimum of 200 characters

in the response. The second, in the form of descriptive words, always asked, "List 3 to 5 words that best describe your current feelings." These questions were inspired by similar language assessments of cognitive, evaluative SWB that have predicted corresponding rating scales at $r$ = .85 [48, 58]. For the language analysis in this paper, we used only the essay. This explains why the number of participants in the language analyses was lower than in the scale analyses.

*Rated momentary, affective subjective well-being.* To assess daily emotions, participants were asked, "How do you feel at this moment?" and rated their emotions on a circumplex affect grid where the y-axis of the affect grid represented *arousal* and the x-axis *valence*, the top of the valence axis representing positive affect [59]. We used valence in the study, which had an ICC of .34, indicating that individuals fluctuated in their day-to-day positive emotions. Two-thirds of the time, participants were asked to rate their *stress* and *burnout*, adopted from the Perceived Stress Scale [60] and Oldenburg Burnout Inventory [61], respectively (Since participants either described their feelings in 3 to 5 words or 2 to 3 sentences and were asked to complete EMA's 3 times a day, stress and burnout-related questions were only asked at the start and end of their day in order to minimize the redundancy of the questions). The stress and burnout questions asked participants to indicate on a five-point scale how much they agree with the statements "I feel nervous and stressed" and "I feel emotionally drained" for stress (ICC = .61) and burnout (ICC = .65), respectively. These ratings were within-centered and combined, which yielded a Cronbach's Alpha of .73 and Mcdonald's Omega of .75, indicating good reliability.

*Drinking.* Participants reported how much they had been drinking the past 24 hours by answering the question, "How many standard drinks did you have yesterday?" along with a picture illustrating standard units, adapted from the National Institute on Alcohol Abuse and Alcoholism [62]. Since the question asked for drinks during the past 24 hours, the untouched within-person analysis of this variable analyzed drinking yesterday in relation to today's emotions. We also analyzed drinking tonight. To do this, we transformed the data such that the emotions reported today, day n, predicted the drinking tonight that was reported tomorrow, day n+1 (since the drinking question asked about drinking the past 24 hours).

For the number of drinks, the mean reported drinks per person per day was 1.86 (SD = 2.05), while the median reported drinks per day was 1.29. This hints at the non-normal distribution of drinking (skewness = 1.89, kurtosis = 5.34). We dealt with this non-normal distribution by Anscombe-transforming the drinking variable, which is a method to stabilize variance [63]. The skewness and kurtosis were reduced to 0.94 and 0.52, respectively, by the transformation, and in our analyses, we only used the Anscombe transformed drinking variable, both within and between individuals.

*Drinking alone.* In the last two waves, we asked participants who reported they had been drinking at least one unit in the past 24 hours "You mentioned you had xx standard drink(s) yesterday. Who were you drinking with?" with the response options "I was drinking alone" and "I was drinking with others". For the drinking alone between person analysis, our main variable was composed of the fraction of drinking days a person drinks alone. We did this to clearly contrast drinking alone with social drinking instead of comparing it with both social drinking and being sober. However, this left some participants' scores being based on only one or two drinking days, with the risk of a poor estimate of how much they actually were drinking alone. To deal with this, only participants with at least three drinking days were included in the drinking-alone analyses.

*Composite subjective well-being.* SWB contains a cognitive, evaluative component, such as life satisfaction, and an affective component, including both the presence of positive and the absence of negative affect [24]. The affective component can be assessed retrospectively in a survey (e.g., "How much inspiration/anger did you experience the past two weeks?"), or

momentary, often in an EMA setting (e.g., "How do you feel at this moment?"), where multiple ratings are averaged across time. These components and ways of assessing SWB can be analyzed separately but also as a composite score, the latter being an approach that has been shown to capture various dimensions of SWB [64, 65].

We used a composite score of SWB in both the between and the within-person analysis. The composite score of SWB for the between-person analysis combined both the composite averaged scores of the momentary, affective SWB, as well as the composite score of the affective and evaluative SWB from the baseline/wave surveys.

The momentary SWB specifically included i) valence, ii) stress, and iii) burnout. The baseline SWB from the survey specifically included i) positive affect, ii) negative affect, iii) life satisfaction, and iv) depression. All these seven variables were z-transformed, and the composite score was calculated in the following way:

SWB = avg(momentary SWB, baseline SWB)
where
momentary SWB = valence—avg(stress, burnout)
and
baseline SWB = avg(positive affect, Cantril Ladder)—avg(negative affect, depression)

Whenever a participant had missing data at some of the variables, we calculated the equation without it. The momentary SWB was used for the within-person analysis. The momentary SWB had a Cronbach's Alpha of .73, whereas the total SWB composite score has a McDonald's Omega of .94 and Cronbach's Alpha of .88. Univariate analyses of all the individual variables, including momentary and baseline SWB, can be found in S1-S3 Tables in S1 Appendix. Although different composite scores of SWB are common [64, 65], this is the first time this specific composite score of SWB has been used (to see how the individual variables correlated to the other study variables, see S1-S3 Tables in S1 Appendix).

## Statistical analysis

All language analysis was performed using the Python package DLATK [66] in a virtual environment at Stony Brook University. The cleaning and rating scales analyses used the R packages tidyverse [67], psych [68], lm.beta [69], esvis [70], plyr [71], arsenal [72], corrplot [73], ggcorrplot [74], jmuOutlier [75], car [76], plotrix [77], PerformanceAnalytics [78], naniar [79], proc [80]. The analyses were done in Visual Studio Code and Rstudio.

The analysis comprised both between and within-person analyses, capturing both differences between people and day-to-day changes within people during the study. The between-person analyses involved aggregating all the ratings and language for each participant. The within-person analysis involved individual mean-centering of all the variables and correlation of these, where the p-values of the mean-centered rating analyses were compared to a permuted distribution of p-values ($N^{permutations}$ = 20000). These within-centered correlations were multilevel model fixed effects implemented stage-wise, equivalent to the standardized beta coefficients from multilevel models implemented with joint mode (the stage-wise approach is accurate as long as the number of participants is at least 30 [81]). We included the joint mode within-centered multilevel models in the supplemental material (S4 Table in S1 Appendix), showing identical standardized beta coefficients as the correlations in Fig 1. We also included two models predicting within-centered SWB from within-centered day-to-day drinking and grand mean-centered AUDIT (S5 Table in S1 Appendix) and four grand-mean-centered multilevel models predicting SWB from day-to-day drinking and AUDIT scores (S6 and S7 Tables in S1 Appendix). Overall, the results of these models aligned with the within-centered fixed effects correlations presented in the manuscript.

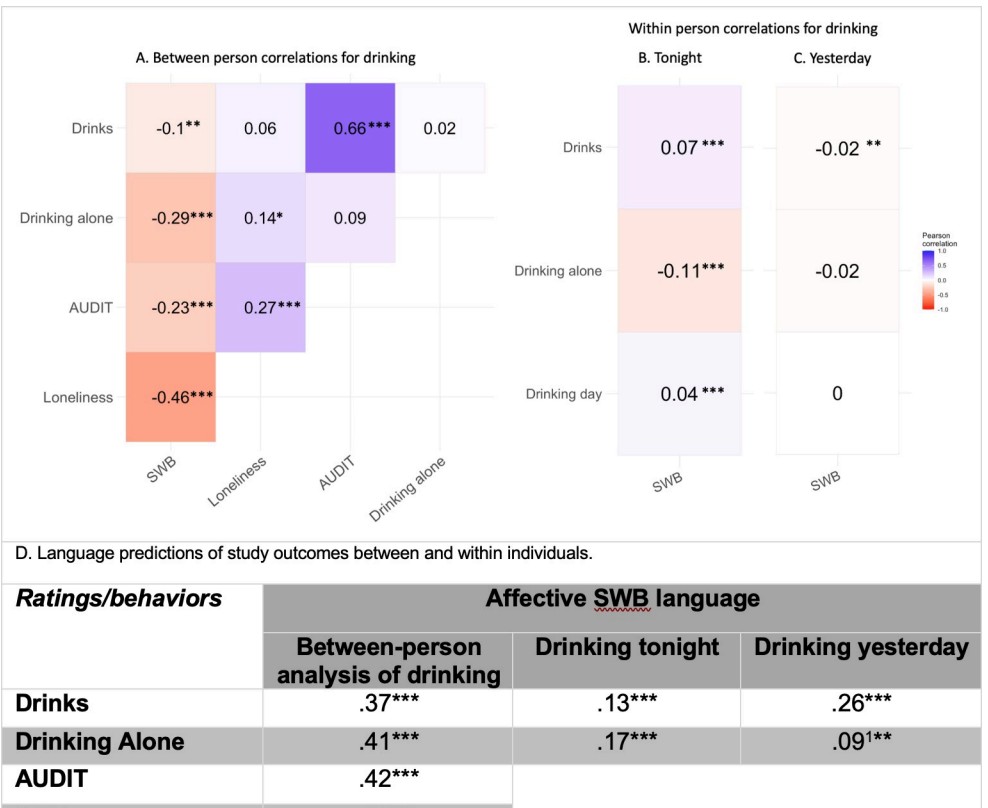

**Fig 1. Correlations of drinking between and within individuals.** *Note*: * p < .05; ** p < .01; *** p < .001. Correlations between drinking variables and subjective well-being (SWB). Correlations within white squares in Fig 1A–1C were non-significant. Drinks were Anscombe transformed. **A**. N = 190–253 for Drinking Alone and Loneliness correlations. N = 891–908 for remaining correlations. **B**. N = 2261 for Drinking Alone correlations. N = 10104 for remaining correlations. **C**. N = 2667 for Drinking Alone correlations. N = 12615 for remaining correlations **D**. Accuracy of models based on language assessments. N = 492 except for Drinking Alone (N = 175) for between-person analyses. For within-person analyses predicting drinking tonight, Drinks (N = 4459), Drinking Alone (N = 1099). For within-person analyses predicting drinking yesterday, Drinks (N = 5675), and Drinking Alone (N = 1315). [1] = feature selection and PCA applied.

We reported Pearson correlations exclusively in this paper for comparison reasons, although standardized beta coefficients would have been equally suitable for the within-person correlations, and logistic regression would have better suited some cases, such as classifying drinking alone days. We did not find substantial deviance when classifying instead of correlating. Whenever comparing two correlation coefficients, we used the Steiger test for comparing dependent correlations [82]. Due to a smaller sample size in the language data, we set the *n* of these dependent comparisons to the mean *n* of the two comparisons. Significance was set to .05.

## Language-based assessment\

For the language analysis, we used a transformers-based large language model [83] to create word embeddings used for predicting study outcomes. Transformers are the state of the art for natural language-based prediction [84], but a downside is their black-box nature and lack of explainability. We, therefore, also created data-driven clusters of co-occurring words called topics that were correlated with the study outcomes to derive language insights [52].

Transformers-based large language models are models trained on large amounts of data from the internet that understand words in their contexts and long-shot dependencies through the transformers technique (see [85]). This means that the models can understand the word "alcohol" differently in the sentences "I love alcohol" and "I hate alcohol". The long-shot dependencies mean that the language model can recall language content further back in the sequence than just a couple of words. We used the large language model Roberta Base [83], layers 11 and 12 (Roberta Base represents each token (i.e., word) with 12 layers with a slightly different semantic value depending on the context, and layers 11 and 12 have shown the best performance in syntactic and semantic structure [86], relevant to human-level tasks), which has performed the best among different large language models for human-level analyses [87]. The model converted the data into word embeddings, which are numerical representations of language supposed to capture the latent meaning of the language [51]. Each language response then gets a vector with, in this case, 1536 numbers. These numbers are features, or dimensions, of the language.

Latent Dirichlet Allocation [52] is an algorithm that computes how n-grams (i.e., $n$ consecutive words; e.g., 1-grams are single words and 2-grams are word pairs) co-occur, ending up in clusters of words called topics, including weights of the words in the topics. We created 200 topics based on the co-occurrences of all the language data in our study based on 1-grams. We excluded the 75 most common words, including words such as "the", and iterated the LDA algorithm 750 times. These topics were then correlated with the study outcomes, with Benjamini-Hochberg [88] corrections for multiple comparisons.

When using the word embeddings as predictors of various outcomes, such as drinking, we applied a ridge regression with 10-fold cross-validation. The features of the word embeddings were the predictor variables. The data was split into 10 folds, each including a random portion of the data, which were further divided into a training and test set. Each fold produced a model based on the training data that was tested on the fold's test data. The final model used the best-performing model and applied it to the entire dataset with one fold (i.e., 10% of the data) as a test set. The presented numbers were the Pearson correlations between the trained models' predicted values and the actual values of the test data. Thus, we used Pearson correlation to evaluate the accuracies of the predictive models. These correlations are theoretically always positive (a negative correlation means that the model is systematically predicting wrongly).

The Roberta Base word embeddings consisted of 1536 numbers (i.e., predictors), and the data on drinking alone were relatively low ($n^{participants}$ = 175 and $n^{EMAdays}$ = 1315), resulting in more predictors than outcomes. We applied univariate feature selection and PCA on most of the drinking alone analysis to deal with this. For the within-person language analysis, the mean-centering of the language data was computed by topic and by feature for the word embeddings.

## Results

The language assessment of SWB was more accurate than SWB ratings in predicting drinking behavior (Fig 1). This was the case both between individuals across multiple weeks (Fig 1A and 1D) and within individuals in their everyday fluctuation of SWB and drinking, both for predicting drinking tonight (Fig 1B and 1D) and yesterday (Fig 1C and 1D). Between individuals, those who drank more reported a significantly lower SWB, but the aggregated language assessments of SWB over several weeks predicted mean drinking with more than thrice the precision ($r$ = -.10 *vs* .37, $p < .001$). Within individuals, days when participants drank more than usual, correlated significantly to a higher SWB, but the language assessment predicted the

drinking with nearly twice the accuracy ($r = .07$ vs $.13$, $p < .001$). The difference was amplified for predicting drinking yesterday ($r = -.03$ vs $.26$, $p < .001$).

The aggregated language assessments of SWB further predicted the AUDIT scores more accurately than SWB ratings did ($r = -.23$ vs $.42$, $p < .001$). These results were similar but weaker and not statistically significant (in correlation difference) when predicting high-risk drinkers ($r = -.21$ vs $.30$, $p = .07$). Both AUDIT scores and mean drinking have a small hump-shaped relationship to SWB (S8 Table in S1 Appendix). The language assessment of SWB showed a strong convergence with rated SWB both between ($r = .69$) and within individuals ($r = .57$, not shown in Fig 1D).

## Drinking alone is negatively related to subjective well-being

Of the 82% of participants who drank at least one day of the study, 66% drank alone at least once. Of all drinking days, 31% occurred alone, and then the participants drank 0.8 fewer drinks than the days they drank with others ($p < .001$). Thus, participants drank significantly more drinks when they drank with others than when they drank alone. The rated SWB of the participants who always drank alone (6%) and always drank with others (29%) deviated significantly from the SWB grand mean negatively and positively, respectively (Table 1). Drinking alone correlated significantly with overall Subjective Loneliness at $r = .14$ (Fig 1A), signaling a small convergence between the loneliness measure and the alone drinking behavior.

On days when individuals drank alone, they reported significantly lower overall SWB ($r = -.11$, Fig 1B), whereas they did not report a significant difference in SWB the day after they had been drinking alone. Between individuals, the fraction of the drinking days participants drank alone predicted SWB more accurately than drinking and AUDIT scores did. The SWB correlation to drinking alone ($r = -.29$, Fig 1A) was nearly three times stronger and significant compared to the SWB correlation to mean drinking ($r = -.10$) and non-significantly stronger than AUDIT scores ($r = -.23$).

When added in the same multiple regression model (Table 2), drinking alone and AUDIT explained unique parts of SWB ($\beta = -.28$ and $-.26$, respectively), whereas mean drinking did not significantly explain any of the model's accuracy. Thus, while mean drinking did not explain any unique variance of SWB between individuals, both AUDIT and drinking alone did, together explaining 13% of the variance in SWB. AUDIT and drinking alone remained significant predictors after adding Subjective Loneliness to the model (S9 Table in the supporting information). Again, the affective SWB language was more accurate than SWB ratings when predicting drinking alone, both between participants ($r = -.29$ vs $.41$), within participants tonight ($r = -.11$ vs $.17$), and within participants yesterday ($r =$ non-significant vs $.09$), although only the difference was significant in the latter comparison.

**Table 1. Descriptive statics of drinking alone distributions.**

| Group | N | Mean subjective well-being (SE) |
|---|---|---|
| Always drinking with others | 88 (29%) | 0.42* (0.15) |
| Always drinking alone | 17 (6%) | -1.24* (0.41) |
| Never drinking | 41 (13%) | -0.04 (0.27) |
| Sometimes drinks alone | 153 (50%) | -0.10 (0.11) |
| Everyone included | 307 | 0.00 (0.08) |

*Note*: *Distribution of drinking alone behavior.* * = $p < .01$, in significance test from the total mean. Only the last two waves asked participants if they drank alone. Thus, there were fewer participants than in the other analyses.

**Table 2. Predicting subjective well-being from drinking behavior.**

| Predictors (β) | | | |
|---|---|---|---|
| **Drinks** | **AUDIT** | **Drinking alone** | **adj R² (r)** |
| - | -.33 | -.26 | .15 (.40) |

*Note*: Multiple linear regression model with AUDIT, drinking alone and drinks predicting subjective well-being between individuals. n = 227. Drinks were Anscombe transformed. Adding sex and age did not add any accuracy to the model, which by themselves had an adj R² of .01. Non-significant = "-".

## Affective subjective well-being language associated negatively with drinking and AUDIT

Drinking related most prominently to language describing social events, whereas problematic drinking (AUDIT) related more to alcohol language (Fig 2). The topics most strongly associated with mean level drinking were a *drinking yesterday* topic ($r = .24$), a *friends dinner* topic ($r = .24$), and a *hangover* topic ($r = .20$). In comparison, problematic drinking (AUDIT) was related to similar topics but stronger to the hangover topic and not to the friends dinner topic. In addition, AUDIT was related to a *drink urge* topic. On days when drinking more than usual, the participants used the *friends dinner* topic more, and also a *friends celebration* and *looking forward* topic. On the days after drinking more than usual, the participants, in addition, used more hangover language (all topics significantly associated with the drinking variables can be found in S5 and S6 Tables in the supporting information). Taken together, the alcohol-associated language involved a strong social aspect, which diminished among problematic AUDIT drinkers.

| Topics |  |  |  |  |  |  |
|---|---|---|---|---|---|---|
| **Topic description** | *Drinking yesterday* | *Friends dinner* | *Hangover* | *Drink urge* | *Friends celebration* | *Looking forward* |
| **AUDIT** | r = .21 | - | r = .23 | r = .17 | - | - |
| **Mean drinking** | r = .24 | r = .24 | r = .20 | - | - | - |
| **Drinking tonight** | - | r = .07 | - | - | r = .08 | r = .05 |
| **Drinking yesterday** | r = .10 | r = .11 | r = .13 | - | r = .10 | - |

**Fig 2. Affective subjective well-being language related to AUDIT and drinking within and between individuals.** *Note*: *Affective language related to drinking behavior, in terms of clusters of similar words, i.e., topics. AUDIT and mean drinking are between-person analyses, and drinking tonight and yesterday are within-person analyses. Word size within each topic represents rank-ordered frequency in the topic. All topics are significant after Benjamin-Hochberg correction for multiple comparisons and controlled for age and gender. $N^{participants}$ = 492 for drinking and AUDIT. $N^{ema\ days}$ = 4459 for drinking tonight and 5675 for drinking yesterday. Non-significant = "-".*

| Topics |  |  |  |  |  |  |  |
|---|---|---|---|---|---|---|---|
| Topic *description* | These surveys | Hate frustration | Always trying | Sad and anxious | Needs and stress | Annoyed and irritated | Nervous |
| **Mean drinking alone** | *r* = .31 | *r* = .30 | *r* = .27 | *r* = .26 | *r* = .26 | - | - |
| **Drinking alone tonight** | - | - | - | - | - | *r* = .10 | *r* = .09 |

**Fig 3. Affective subjective well-being language related to drinking alone within and between individuals.** *Note: Affective language related to drinking alone behavior, in terms of clusters of similar words, i.e., topics. Mean drinking alone is a between-person analysis, and drinking alone tonight is a within-person analysis. Word size within each topic represents rank-ordered frequency in the topic. All topics related to mean drinking alone are significant after Benjamin-Hochberg correction for multiple comparisons and controlled for age and gender. The drinking alone tonight language is only significant without correction for multiple comparisons, but their effect sizes are stronger than most of the within-person correlations of affective language and drinking. $N^{participants}$ = 177 for mean drinking alone. $N^{ema days}$ = 1077 for drinking alone tonight. Non-significant = "-".*

### Affective subjective well-being language associated with drinking alone

Drinking alone was related to language describing being sad, nervous, and annoyed (Fig 3). Five topics were strongly related to drinking alone between individuals, such as the *sad and anxious* (*r* = .29) and *hate frustration* (*r* = .32) topics. On days when drinking alone, individuals used an *annoyed and irritated*, and *nervous* topic more than usual. In general, the language related to drinking alone describes negative aspects of SWB both within and between individuals, converging with the negative correlation between drinking alone and the rated SWB. None of these topics that significantly related to drinking alone were significantly related to drinking or AUDIT. The days following drinking alone involved no significant language and had no significant relationship to reported SWB.

## Discussion

Asking individuals to describe their feelings (i.e., affective, momentary SWB) freely with language is generally more ecologically valid and generally contains more information than asking individuals to rate their SWB with closed-ended rating scales. For all drinking outcomes in this study on US restaurant workers, momentary language assessments of SWB outperformed the rating scales' predictive accuracy (in line with our first hypothesis). This aligns with previous research comparing rating scales and language (e.g., [48, 49]) and is not surprising considering the vast amount of information that language assessments include. This provides further evidence-based support for using language assessments in complement to, or even instead of, rating scales for assessing psychological states and behaviors. The language assessments did not only outperform the rating scales in predictive accuracies of drinking behaviors, but they also provided language-based insights to help understand the mechanisms behind drinking behaviors.

### The social aspect of drinking explains problematic drinking

Both drinking alone and AUDIT captured distinct parts of problematic drinking that were not associated with less problematic drinking. These distinct parts seem to be about the lack of

sociality, which is a key aspect of SWB [6, 34]. The language that related to drinking was consistently social (in line with our third hypothesis), especially on drinking days, when individuals also reported a higher SWB than usual. AUDIT converged strongly with drinking ($r$ = .66, Fig 1), and the affective language related to AUDIT was similar to the language related to drinking, but the social aspect was attenuated for the AUDIT-associated language. In addition, AUDIT was related more strongly to Loneliness, a strong predictor of poor SWB and physical health [36], than to SWB (Fig 1A) but drinking had a non-significant relationship to Loneliness. The fraction of drinking days was even significantly related to higher SWB (S1 Table in S1 Appendix). The multiple regression analysis (Table 2) showed that the small association between drinking and SWB diminished when drinking and AUDIT were simultaneous predictors of SWB. It should be noted that heavy drinkers drive the small correlation between drinking and SWB (S2 Table in S1 Appendix; in line with previous research [28, 29]), and heavy drinking is one of the dominant components of the AUDIT assessment. Likely, the problematic drinking components of AUDIT compose both a heavy drinking and (reduced) social aspect.

Somewhat unexpectedly, AUDIT was distinct from drinking alone in our analysis, both in their correlation (Fig 1A), but also in terms of the language they related to (Figs 2 and 3). Whereas AUDIT scores and drinking were significantly related to similar topics, drinking alone did not relate to any topic that AUDIT or drinking did. The drinking alone-associated language did not even involve alcohol but instead included many aspects related to poor SWB, such as being anxious and sad. The topics also indicated irritation, frustration, and a lack of understanding of their lives (e.g., *hate frustration* and *always trying* topics). This suggests that individuals are indeed drinking alone to cope with negative emotions and internal states, as demonstrated by previous studies (e.g., [38, 40]). Noteworthy, the within-person analyses showed that, in the short run, they succeeded. The drinking alone days were associated with significantly lower SWB, and this relationship diminished the day after drinking alone (Fig 1B and 1C). The opposite pattern was found for drinking, with a higher affect on drinking days but a lower affect following a drinking day (Fig 1B and 1C, and S4 and S5 Tables in S1 Appendix).

Drinking alone and AUDIT were distinct, and previous evidence suggests that drinking alone has been used to predict AUDIT scores 20 years later, beyond drinking and baseline AUDIT scores [33, 45]. The study data indicate that individuals had a lower SWB on days when drinking alone, and they drank alone when describing being nervous and annoyed. Although the negative SWB associated with drinking alone days diminished the following day, repeating this behavior over time might be what leads to problematic drinking in the long run (in line with the longitudinal predictions of AUDIT from drinking alone [33, 45]). If drinking alone is a strong early AUD signal, clinicians should target early interventions to a greater degree for solitary drinkers rather than focusing only on high AUDIT scorers. This would require guidelines for determining problematic drinking alone levels. While there are clear guidelines for determining problematic drinking in terms of AUDIT scores, there are none for problematic drinking alone levels. For example, is any regular alone drinking by adolescents and young adults a major cause for concern, or does a certain level of intoxication need to be reached while drinking alone to indicate a serious problem? Further, a follow-up research question that has received little attention is whether there is any common predecessor of the negative emotions behind drinking alone.

The strong negative association between drinking alone and SWB between individuals also means there is a strong positive relationship between the *fraction of drinking days drinking with others* and SWB ($r$ = .29). Both the presented and previous data [28, 29] show that drinking in low to moderate amounts is associated with the highest SWB. Further, the individuals

who always drank with others in this sample had a significantly higher SWB than the average person. While previous research has experimentally manipulated mood before drinking [41], future research could isolate the drinking aspect of social events to see if alcohol amplifies the social event's positive impact on SWB.

Although the negative consequences of alcohol on physical health are well documented [1–3], more and more evidence suggest that the relationship between psychological health (e.g., SWB) and alcohol consumption is more nuanced, where individuals are happier on drinking days [11, 12] and low to moderate drinkers are often the happiest [28–30]. The language-based insights from the current study underscore that the social aspects of drinking are vital to understanding its relationship with psychological health and SWB. Although alcohol has been documented as a risk for mortality [2], a lack of social relationships has been at least a strong risk factor for mortality [89]. Taken together, the data, in combination with recent studies [11, 12], points toward two clear distinctions of alcohol use for its impact on psychological health and SWB, namely, to what extent it is excessive and to what extent it is social. Both excessive drinking and drinking alone have been documented as to how they decrease psychological and physical health [1–4, 42, 43], whereas fewer studies have pointed towards the potential positive impact that low to moderate social drinking may have on psychological health and SWB.

## Limitations

A trade-off for the ecological nature of this study is the limitation to populations of bartenders and servers being part of professional and online groups. The sample was reflective of the larger population in that it was mostly self-reported females (75%). Further, data was only available for three waves and drinking alone for two waves while the study continued, and more data may be available in the future to power more nuanced analysis. Lastly, though the multi-modal assessments (both self-report and language-based assessments) can mitigate some bias, both language-based assessments and self-report can suffer from social desirability where respondents answer more in line with how they want to be perceived than truthfully, although they were given assurances that they would remain anonymous.

## Conclusion

It is important to distinguish between *alcohol* drinking and *problematic alcohol* drinking, especially in terms of psychological health and SWB. Compared to using rated emotions, this distinction is facilitated and has increased accuracy when utilizing quantitative assessments of individuals' everyday affective language. Our results support that a strong social factor impacts the relationship between drinking and SWB. Both drinking alone and problematic drinking (i.e., AUDIT) are distinct predictors of low SWB, and they both include components of loneliness. This is evident in comparison to drinking, which our data show is generally a very social behavior.

## Supporting information

**S1 Appendix. Supporting information for language-based EMA assessments help understand problematic alcohol consumption.**
(DOCX)

## Author Contributions

**Conceptualization:** August Håkan Nilsson, Hansen Andrew Schwartz, Richard N. Rosenthal, James R. McKay, Lyle Ungar.

**Data curation:** Hansen Andrew Schwartz, Huy Vu, Young-Min Cho, Adithya V. Ganesan.

**Formal analysis:** August Håkan Nilsson.

**Funding acquisition:** Hansen Andrew Schwartz, Richard N. Rosenthal, James R. McKay, Lyle Ungar.

**Investigation:** August Håkan Nilsson, Lyle Ungar.

**Methodology:** August Håkan Nilsson, Richard N. Rosenthal, Huy Vu, Lyle Ungar.

**Project administration:** Hansen Andrew Schwartz, Syeda Mahwish.

**Software:** Adithya V. Ganesan, Lyle Ungar.

**Supervision:** Hansen Andrew Schwartz, Richard N. Rosenthal, Lyle Ungar.

**Validation:** Lyle Ungar.

**Visualization:** August Håkan Nilsson, Lyle Ungar.

**Writing – original draft:** August Håkan Nilsson.

**Writing – review & editing:** August Håkan Nilsson, Hansen Andrew Schwartz, Richard N. Rosenthal, James R. McKay, Syeda Mahwish, Lyle Ungar.

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
