## [Decision Letter · Decision Letter 0]

7 Nov 2023

PONE-D-23-17020Language-based EMA Assessments Help Understand Problematic Alcohol ConsumptionPLOS ONE

Dear Dr. Nilsson,

Thank you for submitting your manuscript to PLOS ONE. After careful consideration, we feel that it has merit but does not fully meet PLOS ONE’s publication criteria as it currently stands. Therefore, we invite you to submit a revised version of the manuscript that addresses the points raised during the review process. The reviewers saw merit in the study, including seeing value in the use of language-based assessment. However, a number of issues were also raised that require addressing before the manuscript is ready for publication. In particular, Reviewer 1 highlighted the significant limitations of separate within-person and between-subjects analyses of data. Multilevel modeling is more appropriate as it can take into account the nested structure of the data. I would also suggest the authors consider PLOS ONE's data availability requirements in their resubmission.

We look forward to receiving your revised manuscript.

Kind regards,

Matthew J. Gullo

Academic Editor

PLOS ONE

Journal Requirements:

"Initials of the authors who received each award: Ungar, L., Schwartz, A.

Grant numbers awarded to each author: 1R01AA028032-01

The full name of each funder. National Institute on Alcohol Abuse and Alcoholism

URL of each funder website: https://www.niaaa.nih.gov/

Did the sponsors or funders play any role in the study design, data collection and analysis, decision to publish, or preparation of the manuscript? No"

Reviewers' comments:

Reviewer's Responses to Questions

**Comments to the Author**

1. Is the manuscript technically sound, and do the data support the conclusions?

Reviewer #1: Partly

Reviewer #2: Yes

2. Has the statistical analysis been performed appropriately and rigorously? 

Reviewer #1: No

Reviewer #2: Yes

3. Have the authors made all data underlying the findings in their manuscript fully available?

Reviewer #1: No

Reviewer #2: Yes

4. Is the manuscript presented in an intelligible fashion and written in standard English?

Reviewer #1: Yes

Reviewer #2: Yes

5. Review Comments to the Author

Reviewer #1: The current paper aimed to determine how affective language reported by restaurant workers is associated with problematic and with solitary alcohol consumption. It also aimed to compare the accuracy of subjective wellbeing language measure to ratings scale measure in predicting drinking behaviour. The authors used ecological momentary assessment method to collect the data. Results showed that language-based assessments were better predictors of drinking compared to scale assessments. The study found that problematic drinking was related to reporting alcohol-related language and solitary consumption. The study also showed that on solitary drinking days, individuals reported lower subjective wellbeing, whereas social consumption days were associated with higher subjective wellbeing.

The premise of this paper is interesting and unique. However, the structure of the paper needs to be improved, as at the moment it is lacking clarity. I also believe that additional statistical analysis will make the paper stronger.

Abstract:

- P.2. “Report their alcohol behavior the past 24 hours”. “In” is missing after “behavior”.

- P. 2. Conclusion: “Being alone moderated this relationship”. There was no moderation analysis conducted in this study, therefore the statement is incorrect.

Introduction:

- The argument made in the introduction – that researchers can learn more about problematic alcohol consumption – is original and interesting. However, introduction needs to include more theory to place the argument in context. The two components of SBW are just briefly mentioned, making it harder for the reader to follow. The rationale behind using SBW, rather than separately looking at affect valence and arousal, is also not explained.

- Rationale for using restaurant workers is not presented.

- Some points made by authors need to be placed in context. For example, “Most drinking occurs socially but 27% of adolescents and 40% of young adults report they have been drinking alone the past 12 months” (p. 4). What population are your referring to here? Similarly, it is unclear what is meant by “Harmony of Life” on p. 4.

- Some claims are not referenced, for example:

o “restaurant workers, a group at increased risk for problematic alcohol use” (p. 3).

o “non-problematic drinking can have a positive relationship to mental health and SWB through its strong social component” (p. 3).

o “individuals drink because of different reasons and because of the hump- shaped relationship drinking has to Life Satisfaction” (p. 5).

- Several key papers on the topic are missing in the introduction. I encourage the authors to consider the following publications in their literature review:

Bresin, K., Mekawi, Y., & Verona, E. (2018). The effect of laboratory manipulations of negative affect on alcohol craving and use: A meta-analysis. Psychology of Addictive Behaviors, 32(6), 617.

Tovmasyan, A., Monk, R. L., & Heim, D. (2022). Towards an affect intensity regulation hypothesis: Systematic review and meta-analyses of the relationship between affective states and alcohol consumption. PloS one, 17(1), e0262670.

Tovmasyan, A., Monk, R. L., Sawicka, I., & Heim, D. (2022). Positive but not negative affect is associated with increased daily drinking likelihood in non-clinical populations: systematic review and meta-analyses. The American Journal of Drug and Alcohol Abuse, 48(4), 382-396.

Weiss, N. H., Kiefer, R., Goncharenko, S., Raudales, A. M., Forkus, S. R., Schick, M. R., & Contractor, A. A. (2022). Emotion regulation and substance use: a meta-analysis. Drug and Alcohol Dependence, 230, 109131.

- Please use past tense when describing the study.

- “More than half of the 38 authors expected drinking to be higher on days with more negative affect”. “Authors” needs to be replaced by “studies”. (p. 3).

- Please avoid using causal language, for example in “high consumption of alcohol can lead to an Alcohol Use Disorder (AUD)”, “can lead” needs to be replaced with “may be associated with” (p. 3). Similarly, “A recently published 17-year longitudinal study confirmed these effects” (p. 4). “Confirmed” needs to be replaced with “supported” or similar.

- “Individuals who drink more alcohol and drink more alone in general” (p. 4) needs to be replaced with “…who drink more alone and who drink more in general”.

- To guide the reader, clear aims, research questions, and hypotheses need to be identified right before the methods section.

Methods:

- Please use past tense throughout.

- Data availability statement is missing.

- “Procedure” heading is missing. Procedure section is lacking necessary details. For example, “participants were notified one to three times a day” (p. 5) –what was it dependent on? “…the text data participants provided at times included sensitive personal data” (p. 5). How was it dealt with? Further, procedure section needs to be structured better, as at the moment it is one big paragraph than needs to be split into two or three.

- “When analyzing Drinking Tonight, a total of 10171 EMA days remained” (p. 5). This is confusing, as there is no description of “drinking tonight” variable up until this point. This later comes up in “statistical analysis section”, but a brief description of what it is needs to be presented here.

- Materials section is not clear. Baseline section should precede EMA section. Each measure, including EMA alcohol measures, need to have its own subheading, there needs to be a description of what exactly does it measure, where was it adapted from (or whether it was developed for this study, and, if so, what was the process), and information on internal consistency, reliability, and validity of the measures.

- “Two-thirds of the time, participants were asked to rate their stress and burnout” (p.6). Why two-thirds of the time?

- “These scales were assessed both at baseline and in each wave survey” (p. 6). This is confusing. How many times did participants have to do baseline survey? Please describe this in the procedure section.

- Part of the “statistical analysis” section belongs in “materials” section.

- Is calculating SWB in such way an established procedure, or was it developer specifically for this study?

- Language-based assessment method will be unfamiliar to some readers, therefore, more details on it are required. For example, “We used the large language model Roberta Base [53], layers 11 and 12” (p. 8) – what are layers 11 and 12? “We created 200 topics based on the co-occurrences of all the language data in our study based on 1-grams” (p. 8) – what does 1-grams refer to?

- The software used for the analyses need to be placed at the beginning of the “analytic strategy” section.

Results:

- I appreciate conducting separate analysis of within- and between-person data. However, I think it would make more sense to conduct multilevel modelling analysis of the available data, as it would be more nuanced and informative.

- “The aggregated language assessments of SWB further predicted the AUDIT scores more accurately than SWB ratings did” (p. 9). The use of word “predict” is inappropriate, as this is correlational analysis.

- “…a strong convergence with rated SWB both between (r = .69) and within individuals (r = .57, not shown)” (p. 9). Why did you decide to not demonstrate .57 correlation in the table?

- “Table 1. Correlations of drinking and between and within individuals” (p. 9). “And” before “between” should be removed.

- Table 1c – I was surprised to see “arousal” as a variable, as it did not come up in the analytic strategy.

- Table 1d – “between” column should be renamed to “Between-person analysis of drinking” or similar.

- Table 2 (p. 10) – SE is missing for most variables.

- P. 10 – full stop is missing when reporting .29 and .10 correlations.

- “r = NS vs .09” (p. 10) – what is NS referring to?

- Please report all correlations in the tables, even if they are not significant.

Discussion:

- Discussion does not address existing theory to show how current findings could take it further.

- I was confused by what is meant by “mere drinking”.

- “AUDIT was distinct from drinking alone in our analysis. Both in their correlation (table 1a), but also in terms of the language they related to (tables 3 and 4)” (p. 13). These two sentences need to be combined into one.

- “repeating this behavior over time might be what leads to problematic drinking in the long run” (p. 13) – please provide references to support this claim.

Other:

- More cautious language needs to be used throughout the manuscript. E.g., “friends who are essential for subjective wellbeing” (p. 2), “the happiest people spend the least time alone” (p. 4), “This is evident in comparison to drinking, which in general is a very social behavior” (p. 14) are definite claims and references to back them up are not included.

Reviewer #2: This is an interesting article providing a novel methodology to yield insights into the effect of alcohol consumption on subjective well-being. The analysis appears to have been performed rigorously and the results are presented clearly. The paper could benefit from some minor revisions to the language to make the text more intelligible, for example:

The following sentence on page 4 is not easy to understand, particularly the end of the sentence. "The happiest people spend the least time alone and in some study samples not shown even one exception of having at least one close relationship [6]."

In the following sentence and throughout the paper, 'all responses' should be changed to 'total responses'. "And second, by analyzing the between-person variation of the aggregated language from each person’s all EMA responses we will be able to have a noise-reduced assessment of the affective SWB language that relates to individuals who drink more alcohol and drink more alone in general."

There may be a better way to phrase the following sentence, as it stands it is quite difficult to interpret, relating to the use of the phrase 'drinking tonight': "We also analyzed Drinking Tonight, then we transformed the data such that the emotions reported today predict the drinking tonight that was reported tomorrow." (page 6)

Although there is a general ethics statement response for the paper guidelines, it might be good to describe in a bit more detail the ethics of recruiting from social media and how this took place.

The paper could do with an overall proofread – for example, 'solitary drinkers' may be better than ‘alone drinkers’.

6. PLOS authors have the option to publish the peer review history of their article (what does this mean?). If published, this will include your full peer review and any attached files.

Reviewer #1: No

Reviewer #2: No

---

## [Author Response · Author response to Decision Letter 0]

30 Nov 2023

Dear editor and reviewers,

Thank you for your valuable feedback. We have now revised the manuscript based on your suggestions and comments. We believe that the comments we have addressed from reviewer 2 and, in particular, reviewer 1 have substantially improved the manuscript’s clarity. We have clarified that our initial results were multilevel model fixed effects implemented stage-wise through within mean-centring and that we used the Pearson correlations for greater interpretability. We have done joint mode multilevel models, and the standardised beta coefficients of these were identical to the Pearson correlations. We have also added the random effects and more mixed models with multiple predictors in the supplemental material as a result of these extra analyses. Further, when re-running the analyses, we discovered that we used the 3-item AUDIT-C and not the 10-item AUDIT in the multiple regression predicting subjective well-being (table 3). Adding the 10-item AUDIT made its std beta coefficient bump from .27 to .33. However, the overall message of this analysis did not change. The non-language data, including open code for the non-language analyses, are now available at Open Science Framework. 

We look forward to hearing from you again.

---

## [Decision Letter · Decision Letter 1]

16 Jan 2024

PONE-D-23-17020R1Language-based EMA Assessments Help Understand Problematic Alcohol ConsumptionPLOS ONE

Dear Dr. Nilsson,

Thank you for submitting your manuscript to PLOS ONE. After careful consideration, we feel that it has merit but does not fully meet PLOS ONE’s publication criteria as it currently stands. Therefore, we invite you to submit a revised version of the manuscript that addresses the points raised during the review process.

Both reviewers commented on the significant improvements in the revised manuscript. Some additional minor comments have been raised by Reviewer 1 that need to be addressed in a subsequent revision.

We look forward to receiving your revised manuscript.

Kind regards,

Matthew J. Gullo

Academic Editor

PLOS ONE

Journal Requirements:

Additional Editor Comments:

Both reviewers commented on the significant improvements in the revised manuscript. Some additional minor comments have been raised by Reviewer 1 that need to be addressed in a subsequent revision.

Reviewers' comments:

Reviewer's Responses to Questions

**Comments to the Author**

1. If the authors have adequately addressed your comments raised in a previous round of review and you feel that this manuscript is now acceptable for publication, you may indicate that here to bypass the “Comments to the Author” section, enter your conflict of interest statement in the “Confidential to Editor” section, and submit your "Accept" recommendation.

Reviewer #1: (No Response)

Reviewer #2: All comments have been addressed

2. Is the manuscript technically sound, and do the data support the conclusions?

Reviewer #1: Yes

Reviewer #2: Yes

3. Has the statistical analysis been performed appropriately and rigorously? 

Reviewer #1: Yes

Reviewer #2: Yes

4. Have the authors made all data underlying the findings in their manuscript fully available?

Reviewer #1: No

Reviewer #2: Yes

5. Is the manuscript presented in an intelligible fashion and written in standard English?

Reviewer #1: Yes

Reviewer #2: Yes

6. Review Comments to the Author

Reviewer #1: Thank you for addressing my comments, the manuscript has been significantly improved. If a few minor points listed below are addressed, I would be happy to recommend this article for publication:

- Abstract: “…friends, who are essential for subjective wellbeing”. As I said previously, more cautious language needs to be used, such as “friends, who could be important for subjective wellbeing”, even if there are references to back up this claim.

- P. 4 - “Most drinking occurs socially” – please reference the claim.

- Method – ethics and data availability sections are usually placed at the beginning of ‘Method’ section.

- Currently, permission is required to access the data on OSF – please remove restrictions to access numerical data and analysis code.

- I would advise to rename ‘Instruments’ to ‘Materials’.

- Make sure r (correlation coefficient) is consistently spelled in italics.

Reviewer #2: I am satisfied that the authors have addressed the comments from the review process and that the manuscript meets the requirements for publication.

7. PLOS authors have the option to publish the peer review history of their article (what does this mean?). If published, this will include your full peer review and any attached files.

Reviewer #1: No

Reviewer #2: No

---

## [Author Response · Author response to Decision Letter 1]

18 Jan 2024

Rebuttal letter

Dear editor and reviewers,

Thank you for your feedback. We have revised the manuscript based on the suggestions from reviewer 1. The non-language data, including open code for the non-language analyses, are now publicly available at Open Science Framework. 

Regarding sources from the “Journal requirements” comment, we double-checked the sources and found one of the articles 

Massin S. Alcohol consumption and happiness: an empirical analysis using Russian panel data. 2011. https://www.academia.edu/2220477/Alcohol_consumption_and_happiness_an_empirical_analysis_using_Russian_panel_data

is a paper published in the research unit The Sorbonne Economics Centre (with no peer-review documentation). The paper used the same data as the published article 

Massin S, Kopp P. Is life satisfaction hump-shaped with alcohol consumption? Evidence from Russian panel data. Addict Behav. 2014 Apr 1;39(4):803–10.

which we refer to similarly in our manuscript. We now removed the non-peer-reviewed reference from the manuscript while double-checking all other references.

Kind regards

August

Journal Requirements:

Reviewer #1: Thank you for addressing my comments, the manuscript has been significantly improved. If a few minor points listed below are addressed, I would be happy to recommend this article for publication:

Thank you for the feedback and for helping us improve the manuscript by addressing these issues.

- Abstract: “…friends, who are essential for subjective wellbeing”. As I said previously, more cautious language needs to be used, such as “friends, who could be important for subjective wellbeing”, even if there are references to back up this claim.

✅ We edited the sentences. It now reads:

“... friends, who often are important for subjective well-being “

- P. 4 - “Most drinking occurs socially” – please reference the claim.

✅ We had the reference at the end of that sentence, but based on your comment, we put it directly after the quotation to clarify the reference: 

“Most drinking occurs socially [34], but 27% of adolescents and 40% of young adults report, in a 4000+ US nationally representative sample, they have been drinking alone at least once during the past 12 months [34].”

- Method – ethics and data availability sections are usually placed at the beginning of ‘Method’ section.

✅ We have edited this!

- Currently, permission is required to access the data on OSF – please remove restrictions to access numerical data and analysis code.

✅ Apologize for this. Now, the code and numerical data are publicly available here.

- I would advise to rename ‘Instruments’ to ‘Materials’.

✅ We have edited this!

- Make sure r (correlation coefficient) is consistently spelled in italics.

✅ Thanks for noticing this. We have edited accordingly!

Reviewer #2: I am satisfied that the authors have addressed the comments from the review process and that the manuscript meets the requirements for publication.

We are glad that you are satisfied with the revision, and thanks for your valuable feedback.

---

## [Editor Report · Decision Letter 2]

23 Jan 2024

Language-based EMA Assessments Help Understand Problematic Alcohol Consumption

PONE-D-23-17020R2

Dear Dr. Nilsson,

We’re pleased to inform you that your manuscript has been judged scientifically suitable for publication and will be formally accepted for publication once it meets all outstanding technical requirements.

Kind regards,

Matthew J. Gullo

Academic Editor

PLOS ONE
---

## [Editor Report · Acceptance letter]

9 Feb 2024

PONE-D-23-17020R2 

PLOS ONE

Dear Dr. Nilsson, 

I'm pleased to inform you that your manuscript has been deemed suitable for publication in PLOS ONE. Congratulations! Your manuscript is now being handed over to our production team.

Kind regards, 

on behalf of

Assoc. Prof. Matthew J. Gullo 

Academic Editor

PLOS ONE